# COVID-19 Pneumonia and Gut Inflammation: The Role of a Mix of Three Probiotic Strains in Reducing Inflammatory Markers and Need for Oxygen Support

**DOI:** 10.3390/jcm11133758

**Published:** 2022-06-28

**Authors:** Angela Saviano, Annalisa Potenza, Valentina Siciliano, Carmine Petruzziello, Claudia Tarli, Alessio Migneco, Flavia Nasella, Francesco Franceschi, Veronica Ojetti

**Affiliations:** 1Emergency Medicine Department, Fondazione Policlinico Universitario A. Gemelli, IRCCS, 00165 Rome, Italy; angela.saviano@policlinicogemelli.it (A.S.); alessio.migneco@policlinicogemelli.it (A.M.); flavia.nasella@gmail.com (F.N.); francesco.franceschi@policlinicogemelli.it (F.F.); 2Internal Medicine Department, Fondazione Policlinico Universitario A. Gemelli, IRCCS, 00165 Rome, Italy; annalisa.potenza@policlinicogemelli.it (A.P.); claudia.tarli@policlinicogemelli.it (C.T.); 3Department of Infectious Diseases, Fondazione Policlinico Universitario A. Gemelli, IRCCS, 00165 Rome, Italy; valentina.siciliano@policlinicogemelli.it; 4Emergency Medicine Department, Ospedale San Carlo di Nancy, 00165 Rome, Italy; carminepetruzziello@live.it

**Keywords:** COVID-19, probiotics, fecal calprotectin, oxygen support, SARS-Cov-2, Lactibiane Iki^®^

## Abstract

**Background:** COVID-19 disease, which typically presents with respiratory symptoms, can trigger intestinal inflammation through SARS-CoV-2 replication in the gastrointestinal tract. Supplementation with probiotics may have beneficial effects on gut inflammation due to their analgesic and anti-inflammatory properties. The primary objective of our study was to evaluate the efficacy of a mix of three probiotic strains (*Bifidobacterium lactis LA 304, Lactobacillus salivarius LA 302,* and *Lactobacillus acidophilus LA 201*; Lactibiane Iki^®^) in the reduction in fecal calprotectin in patients with COVID-19 pneumonia, compared to a control group. The secondary aim was to evaluate the reduction in oxygen support and length of hospital stay in patients taking the probiotic mix. **Patients and Methods**: We conducted a prospective randomized controlled trial at Fondazione Policlinico Gemelli, Rome. We enrolled patients with COVID-19 interstitial pneumonia. One group received the probiotic mix twice a day for 10 days in addition to the standard COVID-19 therapy, and a second group received standard COVID-19 therapy without probiotics. We administered oxygen support (through Ventimask or Optiflow^®^) on days (D) 1, 3, 5, 7 and 10, and the level of fecal calprotectin between D3–D5 and D7–D10. **Results:** A total of 80 patients (44 M/36 F; mean age: 59.8 ± 17.3) were enrolled with a mean value of calprotectin at enrollment of 140 mg/dl. At D7–10, the probiotic group showed a 35% decrease in fecal calprotectin compared to 16% in the control group, a decrease in C-reactive protein (CRP) of 72.7% compared to 62%, and a slight but not significant decrease in oxygen support compared to the control group. **Conclusion:** Supplementation with a mix of probiotics for 10 days in patients with COVID-19 interstitial pneumonia significantly reduces inflammatory markers.

## 1. Introduction

COVID-19 infections have been a global health emergency since 2019 [1,2]. SARS-CoV-2, belonging to the β-Coronavirus group, is mainly responsible for respiratory symptoms such as fever, cough, interstitial pneumonia [3], respiratory failure, and acute respiratory distress syndrome (ARDS) [1,4,5].

The clinical course of the infection can be summarized in three phases. In the first phase, SARS-CoV-2 binds to its ACE2 receptors and then penetrates into the cells of the host and begins replication. In symptomatic cases, this phase is clinically characterized by the presence of mild-to-moderate symptoms (such as headache, sore throat, and loss of taste and smell), fever, and dry cough. In most cases, the immune system can block the infection at this stage and the course is benign. If the disease progresses, it evolves into a second phase, characterized by morpho-functional alterations in the lungs caused by both the cytopathic effects of the virus and the host immune response.

This phase is characterized by a picture of interstitial pneumonia, very often bilateral, associated with respiratory symptomatology that leads to progressive clinical instability with respiratory failure and the need for oxygen therapy. The latter scenario, in a limited number of people, may evolve towards a worsening clinical picture dominated by a cytokine storm and a subsequent hyperinflammatory state. These determine the local and systemic consequences associated with arterial and venous vasculopathy with thrombosis of small vessels and the evolution towards serious and sometimes permanent lung lesions (pulmonary fibrosis).

The final stage of this very serious clinical picture leads to severe ARDS and, in some cases, to the triggering of disseminated intravascular coagulation. In this phase, a progressive alteration of some inflammatory parameters—such as CRP, ferritin, and pro-inflammatory cytokines (interleukin (IL)-2, IL-6, IL-7, IL-10, colony stimulating factors (GSCF), monocyte chemoattractant protein-1 (MCP-1), macrophage-inflammatory protein (MIP)-1A, and tumor necrosis factor (TNF)-α)—and coagulation parameters, including increased levels of fibrin degradation products such as D-dimer, consumption of coagulation factors, and thrombocytopenia, have been observed [5,6,7,8].

Data currently available in the literature show a significant association between the incidence of clinically severe forms of COVID-19 infection and the following conditions: age > 65 years; male sex; tobacco habit; and chronic diseases such as neoplasms, obesity (BMI ≥ 30 kg/m^2^), cerebrovascular diseases, cardiovascular diseases, type I and type II diabetes mellitus, chronic renal failure, and chronic pulmonary diseases [7]. It is unclear why some patients develop a very aggressive disease after 5–7 days and whether any actions can be taken preventively to avoid a fatal evolution.

COVID-19 may also provoke gastrointestinal manifestations such as nausea, abdominal pain, vomiting, and diarrhea [6]. Studies suggest that the virus can infect the gastrointestinal system, replicate in the epithelium of the small and large intestine [6], and produce an excessive immunological reaction [8], resulting in extensive tissue damage. Fecal calprotectin, extensively studied in inflammatory bowel disease (IBD), is a useful tool in defining damage to the intestinal mucosa. A recent trial showed that high levels of fecal calprotectin correlate with a worse prognosis in patients with COVID-19 pneumonia due to the systemic involvement of the infection [9]. A recent paper suggested a crosstalk between the gastrointestinal tract and the respiratory system [10,11].

In this context, supplementation with probiotics may have a beneficial effect on intestinal inflammation, as shown in the literature on IBD with high levels of fecal calprotectin. Probiotics have analgesic and anti-inflammatory properties and can reduce gut inflammation. A mix of three probiotic strains (*Bifidobacterium lactis LA 304, Lactobacillus salivarius LA 302,* and *Lactobacillus acidophilus LA 201*; Lactibiane Iki^®^, Biocure (PiLeJe Groupe), Milan, Italy/PiLeJe Laboratoire, Orée-d’Anjou, France) had significant beneficial effects on inflammation and symptoms of acute uncomplicated diverticulitis [12].

There are currently no data on a potential effect of this probiotic mix in patients with COVID-19 infection and interstitial pneumonia and whether its integration into standard therapy can reduce the severity of clinical manifestations, both pulmonary and gastrointestinal. Therefore, the objective of our study was to evaluate the effects of this mix of three probiotic strains on the reduction in inflammatory markers, abdominal pain, and respiratory symptoms in patients with COVID-19 infection with interstitial pneumonia and increased fecal calprotectin.

## 2. Patients and Methods

Study design and setting

Our clinical study was monocentric and interventional and was conducted at the Fondazione Policlinico Universitario A. Gemelli, IRCCS, Rome, Italy. It is the largest teaching hospital in Rome and a reference center for COVID-19 in central Italy with an average of 80,000 admissions per year. This study was performed with randomization in two parallel and consecutive groups (interventional arm and standard arm). The randomization was performed with an allocation ratio of 1:1 with a Statistical Software STATA 15^®^. In Group A (interventional arm), patients were treated according to the standard of care and supplemented with a mix of three probiotic strains (*Bifidobacterium lactis LA 304, Lactobacillus salivarius LA 302,* and *Lactobacillus acidophilus LA 201*; Lactibiane Iki^®^, Biocure (PiLeJe Groupe), Italy/PiLeJe Laboratoire, France) twice a day for 10 days. Supplementation started on Day 1 (the day of enrollment). In Group B (standard arm; control group), patients were treated according to the standard of care without probiotics. The probiotic mix provided by PiLeJe Laboratoire in a sachet form contains 40 × 10^9^ colony-forming units (CFU), specifically *Bifidobacterium lactis LA 304* at 6 × 10^9^ CFU, *Lactobacillus salivarius LA 302* at 28 × 10^9^ CFU, and *Lactobacillus acidophilus LA 201* at 6 × 10^9^ CFU. One sachet of the probiotic mix was administered 30 min after meals, twice a day in a glass of water: in the morning and in the evening. The product was stored at room temperature (25 °C). Protocol adherence was verified by counting the number of sachets in the boxes returned by the patients the day after finishing the supplementation and by directly questioning the patients about the completion of the supplementation.

2.Eligibility criteria

From February 2021, all patients admitted to our hospital for SARS-CoV-2 infection with respiratory and/or gastrointestinal symptoms and radiological imaging (chest X-ray/CT with/without contrast) of interstitial pneumonia were enrolled. COVID-19 diagnosis was confirmed with a real-time reverse transcriptase polymerase chain reaction assay of nasal and pharyngeal swab specimens. Testing for COVID-19 was conducted according to the World Health Organization (WHO) interim guidance (WHO/2019-nCoV/laboratory/2020.1). Patients with mild COVID-19 disease (requiring oxygen support with 2 L or 4 L nasal cannula or Ventimask 28–35–40% to reach a SpO_2_-target of 95%) were enrolled. We excluded patients on antibiotic therapy or patients who had taken antibiotics in the previous month; patients with IBD, IBS, or infectious colitis of non-COVID-19 bacterial and/or viral etiology; patients with colonic neoplasia (first to last stage); patients who had undergone colon surgery; patients with severe hepatopathy, nephropathy, heart disease or terminal oncological disease; pregnant women; and patients in circulatory shock and under positive pressure support on admission to hospital. All patients gave verbal consent to participate in the study in the presence of two independent witnesses, as the patients themselves could not sign the documents, due to the risk of contamination of paper records.

3.Data collection and measurements

A stool sample was collected for each enrolled patient and was analyzed for fecal calprotectin at patient arrival, and after 5 and 10 days. All patients enrolled in the trial (probiotics group and control group) received treatment with steroids, proton pump inhibitors, heparin, and NSAIDs. Patients received blood tests to assess their white blood cell level and C-reactive protein (CRP) level, as markers of inflammation. The data extracted included age, sex, clinical history and presentation, temperature, heart rate (HR), respiratory rate (RR), blood pressure (BP), Glasgow Coma Scale (GCS) score, oxygen therapy modulated to reach a SpO_2_ target of 95%, and peripheral oxygen saturation (SpO_2_). All clinical signs, including SpO2, and arterial blood gas were measured upon arrival at the Emergency Department (ED). 

4.Analysis of fecal calprotectin

The stool samples were analyzed using a “*LIAISON^®^ Calprotectin*” device to measure the level of fecal calprotectin. This device adopts a chemiluminescence technology (CLIA) with a paramagnetic microparticle solid phase. The assay is a quantitative sandwich immunoassay in which captured antibodies, immobilized to the paramagnetic solid phase, bind calprotectin from the fecal stool samples. The washing eliminates non-specific interactions, allowing for specific detection by conjugate antibodies against calprotectin. The measurement of light from the bound obtained by conjugated isoluminol is in proportion to the quantity of bound fecal calprotectin. The “*LIAISON^®^ Calprotectin*” device includes incubations, washing, calibration, measurement, and analysis. The light signal is proportional to the level of calprotectin present in the samples. A value ≤ 50 µg/g was considered to be normal.

5.Outcomes

The primary outcome of the study was to evaluate the efficacy of a mix of three probiotic strains in patients with COVID-19 infection (and concomitant interstitial pneumonia and intestinal inflammation) in the reduction in inflammatory markers (fecal calprotectin and CRP) compared to a control group with similar characteristics that did not receive the probiotic mix. The secondary outcome was to evaluate any reduction in the need for nasal cannula or Ventimask oxygen support in patients taking the probiotic mix compared to the control group and to evaluate any reduction in the length of hospital stay in these patients. 

6.Sample Size

According to available data [9], up to 50% of patients with COVID-19 may have gastrointestinal involvement. As no data on mean calprotectin values in the COVID-19 population were available in the literature, we assumed a fecal calprotectin level that doubles in these patients (in agreement with what occurs in other diarrheas of viral origin). Considering that the proposed treatment can lower the calprotectin levels in at least 85% of affected cases to a standard value (approximately 50 ± 50 mcg/g in the general population with no symptoms) and no data are available on reference values in the COVID-19 population, a sample of 54 patients in total (COVID-19 positive with intestinal involvement) was needed to achieve 80% power (α of 0.05 using a two-way error test).

7.Statistical analysis

Continuous parametric variables were expressed as mean ± standard deviation, nonparametric variables were expressed as median (interquartile range), and the Mann–Whitney test was used for comparison. Categorical variables were presented as absolute numbers (%) and compared with the Chi2 test (with Yates continuity correction or Fisher’s test, where appropriate). Values of *p* < 0.05 (two-way test) were considered significant. Data were analyzed using SPSS for Windows™ version 25.0 software (IBM, Armonk, NY, USA). 

8.Ethical approval

This study was approved by the local Ethics Committee (authorization #ID3786) and has been performed in accordance with the ethical standards established in the 1964 Declaration of Helsinki and its subsequent amendments.

## 3. Results

### 3.1. Demographic Characteristics

A total of 95 patients were screened for eligibility. Of the 95 patients, 80 (44 M/36 F; mean age: 59.8 ± 17.3) completed the study: 40 (21 M/19 F) for Group A and 40 (23 M/17 F) for Group B (Figure 1). There were no statistically significant differences in age, gender (males and females of both groups), and grade of initial inflammation (mean values of fecal calprotectin and CRP at enrollment) between the two groups (Table 1). The two groups presented similar comorbidities such as hypertension, diabetes, obesity, and chronic obstructive pulmonary disease. None of the enrolled patients had referred acute abdominal pain or diarrhea. All patients were hospitalized 5–7 days after the onset of COVID-19 infection. The prevalence of vaccinated patients was 7 out of 40 in Group A and 8 out of 40 in Group B. During the supplementation period, none of the patients experienced any adverse events. All patients were well informed of the importance of taking the supplementation and took more than 95% of the prescribed doses during the 10 days of supplementation. Dropouts (15/95) were observed as reported in Figure 1, six before the allocation and nine after the allocation. The most common referred symptoms were fever (75/80, 93.7%), dry cough (55/80, 68.7%), dyspnea (53/80, 66.2%), and weakness (42/80, 52.5%), and the mean value of PaO_2_/FiO_2_ obtained performing an arterial blood gas in the emergency room was 327 ± 112 unit. In particular, 36/40 patients in group A showed fever, 23/40 dry cough, 27/40 dyspnea, and 19/40 weakness; for group B, 39/40 patients showed fever, 32/40 dry cough, 26/40 dyspnea, and 23/40 weakness. Pneumonia was diagnosed by chest X-ray in all patients. Both lungs were involved in 38/80 (47.5%) patients.

### 3.2. Inflammatory Markers (Fecal Calprotectin and CRP)

At enrollment, the level of fecal calprotectin was overlapping between the two groups (148.6 ± 50 mg/dl in Group A vs. 135.3 ± 47 mg/dl in Group B, *p* = 0.48 (Figure 2). On Days 3–5, the mean value of fecal calprotectin was 191.8 mg/dl in Group A, whereas in the control group, the mean value was significantly higher (404.04 ± 150 mg/dl; *p* = 0.005). Between enrollment and Days 3–5, the increases in the inflammatory marker level were 29% in Group A and 199% in Group B. On Days 7–10, the mean value of fecal calprotectin decreased to 124.9 ± 46 mg/dl in Group A, compared to 339.0 ± 102 mg/dl in Group B (*p* = 0.006). From Days 3–5 to Days 7–10, the decreases in the inflammatory marker level were 35% in Group A and only 16% in Group B. The CRP level at enrollment was similar between the two groups (66.44 ± 12 mg/dl in Group A versus 71.90 ± 14 mg/dl in Group B, *p* = ns) (Figure 3). On Days 3–5, CRP decreased to a mean level of 18 ± 5 mg/dl in Group A, compared to a mean level of 27 ± 7 mg/dl in Group B (*p* < 0.001), corresponding to a decrease of 72.7% in Group A and 62% in Group B. On Days 7–10, CRP decreased to a mean level of 5 ± 3 mg/dl in Group A, compared to a mean level of 9 ± 2 mg/dl in Group B (*p* < 0.05). We observed a normal white blood cell count in the two groups (6800 ± 1270/mm^3^ vs. 7000 ± 1190/mm^3^; normal value 4000–10,000/mm^3^) at enrollment without significant changes between groups during the period of the study.

### 3.3. Oxygen Support

A slight difference in oxygen support was observed at enrollment, with mean values of 31% in Group A and 28% in Group B (*p* = 0.56) achieving at least a target of SpO_2_ of 95% in both groups (Figure 4). Figure 3 shows the evolution of oxygen demand in the two groups at enrollment and on Days 3, 5, 7, and 10. In Group A, there was a more rapid and continuous reduction in the need for O_2_ support, whereas in Group B, the decrease was smaller. Four patients (10%) from Group B required oxygen support at a level of 60% with high-flow nasal cannula and/or positive pressure mechanical ventilation between three and six days after enrollment; and three patients (7.5%) from Group B were admitted for at least 24 h in an Intensive Care Unit (ICU) during their hospitalization. After a 60-day follow-up period, one (1.25%) patient in Group B died. No patients in Group A required oxygen support at a level of 60% or more. 

### 3.4. Hospitalization Duration

The mean length of hospitalization was 14 ± 6 days in Group A, while in Group B, it was 19.0 ± 10 days (*p* = 0.52).

## 4. Discussion

This is the first interventional study to test the mix of *Bifidobacterium lactis LA 304, Lactobacillus salivarius LA 302, and Lactobacillus acidophilus LA 201* in patients with COVID-19 pneumonia admitted to a hospital. This study confirms the significant correlation between COVID-19 interstitial pneumonia and high levels of fecal calprotectin, associated with a worse prognosis due to the systemic involvement of the infection, regardless of the presence of gastrointestinal manifestations, as demonstrated previously [9]. Studies in the literature suggest that the virus actively infects not only the pulmonary district but also the gastrointestinal district, replicating in the epithelium of the small and large intestine [6,7] and producing an excessive immunological reaction in the host [8], with consequent extensive tissue damage, and in the gut, as confirmed by the increase in fecal calprotectin. In this context, it becomes essential to choose therapeutic measures that can promote viral clearance; reduce pulmonary and intestinal replication and inflammation by decreasing the level of fecal calprotectin; and consequently, improve COVID-19 infection, respiratory symptoms, and patient outcome. 

Many studies [13,14,15,16] indicate that supplementation with probiotics may have beneficial effects on intestinal inflammation due to their analgesic and anti-inflammatory potential and their ability to reduce markers of inflammation (as fecal calprotectin) and abdominal pain. Gutierrez-Castrellon et al. [15] conducted a randomized, quadruple-blinded, placebo-controlled trial, showing that supplementation with a mix of four probiotic strains (*Lactiplantibacillus plantarum KABP033, L. plantarum KABP022, L. plantarum KABP023*, and *Pediococcus acidi-lactici*) was associated with a significant increase in complete viral clearance and remission of symptoms in COVID-19 outpatients after 30 days of supplementation. Patients of the probiotic group reported fewer days of fever, cough, and dyspnea and also fewer days of abdominal pain and diarrhea. The authors showed that the nasopharyngeal viral load on days 15 and 30 was lower in the probiotic group than in the control group and that the improvement in chest X-ray (reduction in mild pulmonary infiltrates) was greater for patients taking probiotics. No adverse effects were reported. Our study showed that patients with COVID-19 infections treated with the mix of probiotics had a more rapid and continuous reduction in the need for oxygen support compared to the control group. The oxygen requirement reduction curve is in fact slightly faster than in the control group, but it does not reach a statistically significant difference. 

Other studies, for example, Foligne et al. [16], demonstrated how the probiotics *Lactobacillus salivarius Ls33* and *Lactobacillus acidophilus NCFM* were able to induce the production of anti-inflammatory interleukins such as IL-10 and reduce pro-inflammatory interleukins such as IL-12, thereby improving intestinal health. 

COVID-19 disease, in its severe presentation, is characterized by a “cytokine storm” with a huge production of pro-inflammatory factors, and lactobacilli could represent a useful tool for mitigating this phenomenon with beneficial effects on health. In particular, SARS-Cov-2 can infect the gastrointestinal system, replicating in the epithelium of the small and large intestine [6] and producing an excessive immunological reaction [8], resulting in extensive gut inflammation and intestinal mucosa damage witnessed by the increased level of fecal calprotectin. Fecal calprotectin is a sensitive marker of gut inflammation. Lactobacilli could play both a direct and an indirect anti-inflammatory role in the intestine, inhibiting pro-inflammatory cytokines, restoring intestinal homeostasis, influencing the induction of T-reg cells, and reducing both local gut inflammation [11] and systemic inflammation (especially in the context of COVID-19 infection that is characterized by pulmonary and extra pulmonary manifestations). Li et al. [17] showed that some probiotics such as *L. acidophilus* and *B. animalis subsp. lactis* exerted a potent anti-inflammatory effect modulating TLR2-mediated NF-κB and MAPK signaling pathways in gut inflammation. In our study, patients treated with probiotics showed a reduction in fecal calprotectin from 148.6 mg/dl to 124.9 mg/dl, while in the control group, the level of calprotectin increased, moving from 135.3 to 339.0 mg/dl after 10 days. 

A limitation of our study was the sample size of patients enrolled (40 patients for each group) and that no data were collected about changes in gut microbiota composition; however, data in the literature underline the positive effects of lactobacilli on gut microbiota composition and balance. In addition, some research [13,14,15,16] indicates that probiotics can act on the immune system by stimulating the action of peripheral lymphocytes, which may exert beneficial effects in COVID-19 disease characterized, conversely, by a status of lymphopenia. Lymphocytes activated against viruses contribute to fighting infections and to restricting viral replication, thereby reducing organ damage. Further investigations are needed to explore this issue.

## 5. Conclusions

Our randomized controlled trial (RCT) showed that supplementation with a mix of three probiotic strains in patients with COVID-19 interstitial pneumonia and intestinal inflammation significantly reduced gut inflammatory markers. Other trials with a larger number of patients are needed to confirm these data.

## Figures and Tables

**Figure 1 jcm-11-03758-f001:**
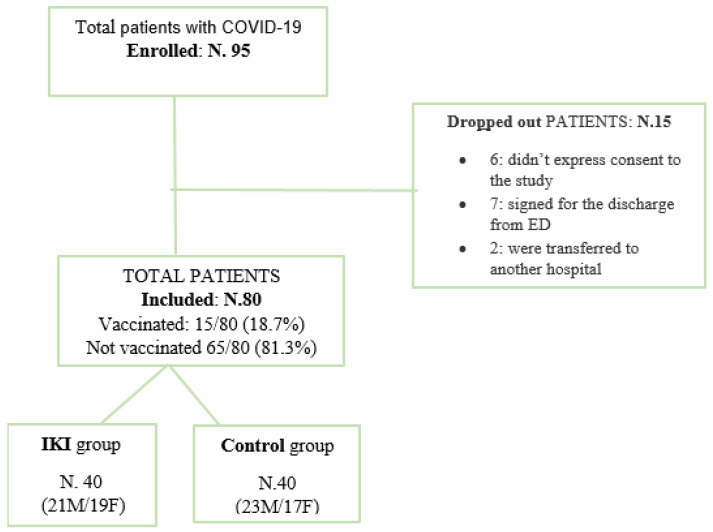
Flowchart of patients’ enrolled.

**Figure 2 jcm-11-03758-f002:**
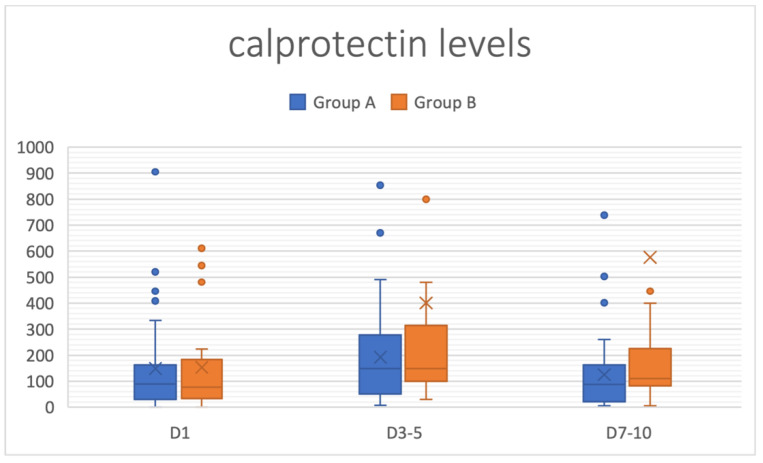
Evolution of mean levels of fecal calprotectin in Group A (supplemented with the probiotic mix) and Group B (control group; standard treatment without probiotics) from enrolment to Days 3–5 and Days 7–10.

**Figure 3 jcm-11-03758-f003:**
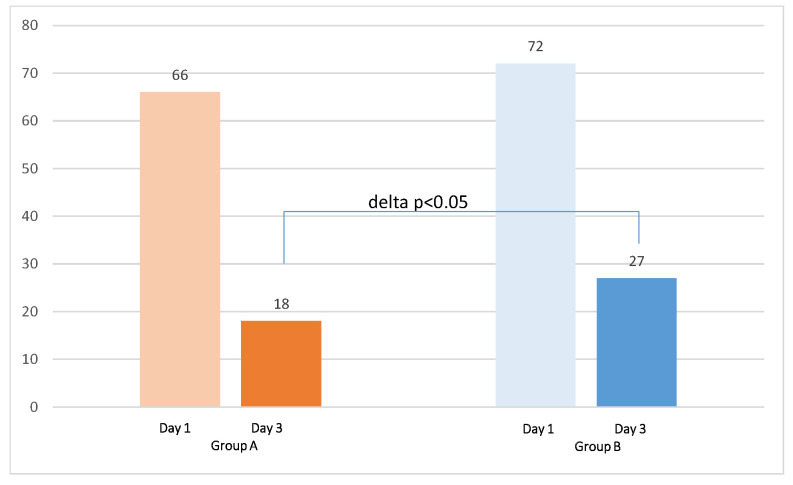
Comparison of mean C-reactive protein (CRP) levels between Group A (supplemented with the probiotic mix) and Group B (control group; standard treatment without probiotics) at enrolment (Day 1) and Day 3.

**Figure 4 jcm-11-03758-f004:**
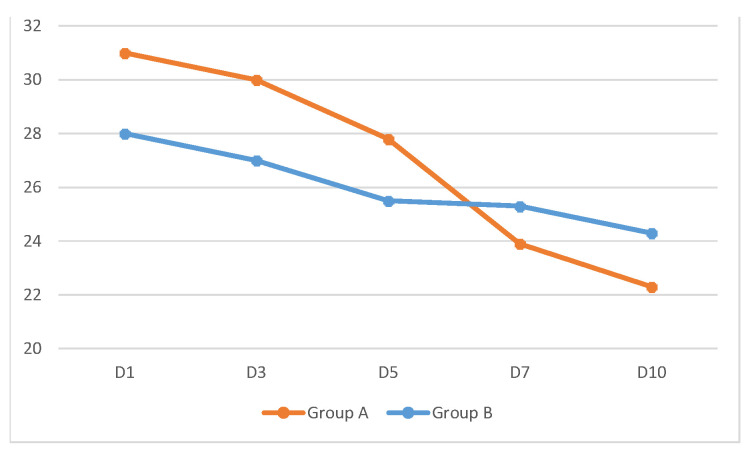
Evolution of mean levels in Group A (supplemented with the probiotic mix) and Group B (control group; standard treatment without probiotics) from enrolment (Day 1) to Day 3, 5, 7 and 10 days.

**Table 1 jcm-11-03758-t001:** Characteristic of patients at baseline (ns: not significant).

	Probiotic Group	Control Group	*p*
Number of patients (M/F)	40 (21/19)	40 (23/17)	ns
Mean Age (years)	59.2 ± 17.8	60.1 ± 15.2	ns
Delta of Fecal Calprotectin (mean value)	148.6 ± 50 mg/dL	135.3 ± 47 mg/dL	ns
Oxygen support (mean FiO_2_)	31%	28.1%	ns
Vaccinated (%)	7/40 (17.5%)	8/40 (20%)	ns
Hypertension	15/40 (37%)	16/40 (40%)	ns
Diabetes	6/40 (15%)	8/40 (20%)	ns
Chronic obstructive pulmonary disease	3/40 (7.5%)	4/40 (10%)	ns
CRP mg/dL	66.44	71.90	ns
D-Dimer	1308	1346	ns
White blood cells	6800/mm^3^	7000/mm^3^	ns

## Data Availability

The data presented in this study are available from the corresponding author upon request.

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
