# Peer review of "COVID-19 Pneumonia and Gut Inflammation: The Role of a Mix of Three Probiotic Strains in Reducing Inflammatory Markers and Need for Oxygen Support"

_jcm, 2022, doi:10.3390/jcm11133758_

Round 1
Reviewer 1 Report
The authors examined the efficacy of a mix of 3 probiotic strains (Bifidobacterium lactis LA 304, Lactobacillus salivarius LA 302, Lactobacillus acidophilus LA 201; Lactibiane Iki®) twice a day for ten days in the reduction of fecal calprotectin in 40 patients with COVID-19 pneumonia, compared to a control group (40 patients). They also evaluated the reduction of oxygen support and length of hospital stay in patients taking the probiotic mix. They found that the probiotic group showed a 35% decrease in fecal calprotectin compared to 16% of the control group, a decrease of C-reactive protein (CRP) of 72.7% compared to 62%, and a more rapid and continuous decrease in oxygen support compared to the control group with a significant improvement in general condition, reduced need for ICU admission, and a more rapid hospital discharge. The findings are interesting, and it helps for novel insight into the beneficial effects of supplementation with probiotics in patients with COVID-19 interstitial pneumonia.
​​MAJOR Comments
Can you compare the COVID-19 vaccine rates between the two groups in Table 1?
I recommend that the author could provide detail about patient status between two groups, e.g., fever, dry cough, dyspnea, weakness, and et. al..
Could you provide any mechanism for a better understanding of your findings?
Please provide the limitations of your study in the Discussion section.
MINOR Comments
Lines 63-67: Could you provide some references for these inflammatory parameters?
Lines 63-67: Please provide the whole term for the inflammatory parameters (e.g., interleukin (IL)-2).
Lines 68-72: The authors only mentioned that data currently available in the literature have shown a significant association, but they didn’t provide any reference.
Line 124: Please make these “SARS-CoV-2” consistent.
Line 218: Please provide the exact p-value; 0.05 could be non-significance.
Lines 225 and 235: Please provide the exact p-value, not only “ns”.
Line 315: Please provide the abbreviation of “RCT” for the first time in your main text.
Author Response
Can you compare the COVID-19 vaccine rates between the two groups in Table 1?
There is no statistical difference between the two group ( p = ns showed in the table)
I recommend that the author could provide detail about patient status between two groups, e.g., fever, dry cough, dyspnea, weakness, and et. al.
We provided
Could you provide any mechanism for a better understanding of your findings?
We provided
Please provide the limitations of your study in the Discussion section.
We modified
MINOR Comments
Lines 63-67: Could you provide some references for these inflammatory parameters?
We provide references
Lines 63-67: Please provide the whole term for the inflammatory parameters (e.g., interleukin (IL)-2).
We modified
Lines 68-72: The authors only mentioned that data currently available in the literature have shown a significant association, but they didn’t provide any reference.
We added reference
Line 124: Please make these “SARS-CoV-2” consistent.
We modified
Line 218: Please provide the exact p-value; 0.05 could be non-significance.
We modified
Lines 225 and 235: Please provide the exact p-value, not only “ns”.
We modified
Line 315: Please provide the abbreviation of “RCT” for the first time in your main text.
We modified
Reviewer 2 Report
The authors studied the probiotics supplementation on reducing inflammatory markers and need for oxygen support. The structure and design of the experiment are clear and simple and the manuscript is presented well.
Some minor modification should be done.
1. Calprotectin is very important parameter in this study, but the analysis method is not clearly described.
2. All bacterial scientific names should be in italic formate. Line 116, 117, 295.
3. Line 305, Lactobacilli is not a scientific name, first letter should not be in capital.
4. The power of the cell number should be in superscript. Line 115, 116 and 117.
5. Fig 1 should be clearly described. How many people are screened for eligibility? And the dropped out were happened before or after allocation, and how many subjects in each group?
6. In line 208, the authors described no dropouts were observed. But in Fig 1, 15 dropouts was mentioned.
7. Fig 2-4, no units in Y axis. And please also mention how the data were presented in each Fig.
8. Fig 2. The scale should be adjusted. Some data are out of the figure.
Author Response
- Calprotectin is very important parameter in this study, but the analysis method is not clearly described.
We clarified
- All bacterial scientific names should be in italic formate. Line 116, 117, 295.
We modified
- Line 305, Lactobacilli is not a scientific name, first letter should not be in capital.
We modified
- The power of the cell number should be in superscript. Line 115, 116 and 117.
We modified
- Fig 1 should be clearly described. How many people are screened for eligibility? And the dropped out were happened before or after allocation, and how many subjects in each group?
We clarified
- In line 208, the authors described no dropouts were observed. But in Fig 1, 15 dropouts were mentioned.
We modified
- Fig 2-4, no units in Y axis. And please also mention how the data were presented in each Fig.
Unit of the Y axis added.
- Fig 2. The scale should be adjusted. Some data are out of the figure.
We modified
Round 2
Reviewer 1 Report
The authors revised the manuscript based on the comments from reviewers. This revised manuscript is acceptable.
Author Response
we fixed the manuscript as suggested
